# Monolithically-stacked thin-film solid-state batteries

Moritz H. Futscher [1,2✉], Luc Brinkman[1,2], André Müller [1], Joel Casella [1], Abdessalem Aribia[1] & Yaroslav E. Romanyuk [1✉]

The power capability of Li-ion batteries has become increasingly limiting for the electrification of transport on land and in the air. The specific power of Li-ion batteries is restricted to a few thousand W kg$^{-1}$ due to the required cathode thickness of a few tens of micrometers. We present a design of monolithically-stacked thin-film cells that has the potential to increase the power ten-fold. We demonstrate an experimental proof-of-concept consisting of two monolithically stacked thin-film cells. Each cell consists of a silicon anode, a solid-oxide electrolyte, and a lithium cobalt oxide cathode. The battery can be cycled for more than 300 cycles between 6 and 8 V. Using a thermo-electric model, we predict that stacked thin-film batteries can achieve specific energies >250 Wh kg$^{-1}$ at C-rates above 60, resulting in a specific power of tens of kW kg$^{-1}$ needed for high-end applications such as drones, robots, and electric vertical take-off and landing aircrafts.

[1] Laboratory for Thin Films and Photovoltaics, Empa—Swiss Federal Laboratories for Materials Science and Technology, Überlandstrasse 129, 8600 Dübendorf, Switzerland. [2] These authors contributed equally: Moritz H. Futscher, Luc Brinkman. ✉email: moritz.futscher@empa.ch; yaroslav.romanyuk@empa.ch

mproving the performance of electrochemical energy storage devices is critical for the electrification of transport on a large scale. The specific energy of Li-ion batteries has improved greatly in recent decades, with record values >500 Wh kg$^{-1}$—close to predicted specific energy limits[1–4]. These batteries employ organic liquid electrolytes and composite cathodes with active electrode particles of ten micrometers in diameter. While such composite cathodes can provide high energies, the long diffusion path of Li ions within the composite cathode limits their rate capability, and thus their specific power[5] with record values up to 2.6 kW kg$^{-1}$ [6]. In addition, capacity fading and Li dendrite growth at high rates is an ongoing challenge and poses a significant limitation and safety risk when combined with flammable organic liquid electrolytes[7–9].

Solid-state batteries (SSBs) have attracted great interest for their ability to increase safety at high charging/discharging rates[10]. However, the specific energy and power of SSBs lag behind that of conventional Li-ion batteries because current solid separators are significantly thicker than the separators used in Li-ion batteries with organic liquid electrolytes[11,12].

Vacuum-based methods enable the fabrication of thin-film SSBs with electrode thicknesses on the order of micrometers in combination with (sub)micrometers-thick separators and current collectors[13,14]. In contrast to wet-chemical methods, solid-state thin films prepared by vacuum-based methods such as sputtering are dense and homogeneous and offer precise control of film thickness and composition[15]. Due to the short diffusion paths, thin-film SSBs offer more than ten times higher charging/discharging rates than composite cathodes[16,17], enabling high powers yet with limited energy due to low areal capacities.

This work presents how a monolithic stack of thin-film cells can enable SSBs with both high energy and power. We demonstrate a prototype of a monolithically (bipolar) stacked thin-film battery with two cells electrically connected in series. Moreover, we predict the specific energy and power of monolithic stacked thin-film batteries using a thermo-electric model. We show that monolithically stacked batteries can potentially achieve specific energies >250 Wh kg$^{-1}$ at charge/discharge times of less than 1 min, resulting in high specific powers of tens of kW kg$^{-1}$. These proposed batteries thus close the gap between supercapacitors and Li-ion batteries and facilitate the electrification of high-end applications such as drones, robots, and electric vertical take-off and landing aircraft (eVTOLs).

## Results

**Stacked thin-film batteries.** All-solid-state thin-film battery cells consist of a vacuum-processed cathode, solid electrolyte, and Li-metal anode, as illustrated in Fig. 1a. The most commonly used solid electrolyte in thin-film cells is Lipon, enabling Li-metal anodes and high-voltage cathodes due to its wide electrochemical stability window from 0 to 5 V vs. Li/Li$^{+}$[18]. Thin-film cells using Lipon were shown to cycle for more than 10,000 cycles with a remarkably low degradation (<10%)[19]—a performance that has not been matched by any other SBB to date. In addition, thin-film cells offer fast charging (<1 min), are non-flammable, have excellent temperature stability suitable for temperature ranges between −40 and 150 °C, and have a meager self-discharge rate of <1% per year[13]. However, thin-film cells have a low areal capacity and are therefore limited to applications with low energy requirements, such as smart cards, medical devices, and small sensors for internet-of-things applications[20].

To improve the energy of vacuum-deposited batteries, one can increase the cathode (and anode) thickness to a few tens of μm (Fig. 1b). However, as the cathode's thickness increases, the cell's current and resistance increase, reducing the power due to energy

lost as heat during cycling[21,22]. For cathode thicknesses greater than a few tens of μm, additives in the cathode are required to ensure sufficient ionic and electronic conductivity, decreasing the cathode's active-to-passive mass ratio.

To increase both the energy and power of vacuum-processed batteries, one can stack several cells on top of each other on a single substrate to form a battery. Monolithic stacking enables the fabrication of stacked thin-film batteries, separated only by thin vacuum-deposited current collectors. The individual cells can be electrically connected in series or parallel (Fig. 1c, d). When the cells are connected in series, the voltages of the stacked cells add up, while the capacity is limited by the cell with the lowest capacity. Such series-connected cells are often referred to as bipolar batteries. When the cells are connected in parallel, the capacities of the cells add up while the voltage is limited by the cell with the lowest voltage. Depending on the application, a high voltage (connected in series) or a high capacity (connected in parallel) design may be advantageous.

Table 1 compares the characteristics of the different vacuum-deposited device configurations. Cells with thick cathodes are suitable for high-energy cells but have a limited rate capability. In contrast, stacked thin-film batteries exhibit low heat losses, resulting in a good rate capability that enables increased power with reduced thermal management requirements. Although stacked thin-film batteries connected in series and parallel have different voltages and capacities, the usable energy and power are the same. Nevertheless, the different connection schemes have implications for the design and operation of the cells. For example, a high battery voltage helps to reduce cable power losses. The series connection further allows for a simpler cell design by eliminating the need for external connections such as tabs and wires. Differences also exist when a cell develops a shunt, e.g., by forming a dendrite. In the case of the parallel-stacked thin-film battery, a shunt in one cell can significantly reduce the effective output voltage. This is in contrast to a series-stacked thin-film battery, where a shunt in one of the cells would reduce the cumulative voltage of the battery. The failure of a cell within the battery stack is thus easier to identify in the series-stacked battery compared to the parallel-stacked battery. In addition, the current flow through the short-circuited cell will be higher in the case of a parallel-connected battery pack, which can locally heat the cell and potentially lead to accelerated degradation of the battery.

We note that stacking cells is a concept also used in conventional SSBs to simplify cell design by reducing external connections and cooling system requirements[23–25]. While bipolar stacked SSBs promise a 30% increase in specific energy and power[26], their rate capability would still be limited by the long diffusion paths within the electrodes. Stacked thin-film batteries have further been discussed in patents[27,28] and previously attempted to be commercialized by a company called Sakti3[29]. However, to our knowledge, there has been no reported demonstration of a working stacked thin-film battery within both patents and academic literature, except for a study published in 2003 which demonstrated only one cycle at a C-rate of C/3[30]. Our work demonstrates thin-film batteries over many cycles with effective C-rates up to 60.

**Experimental proof of concept.** To demonstrate an experimental proof-of-concept of a monolithically-stacked device, we fabricated a (bipolar) stacked thin-film battery consisting of two cells electrically connected in series. Each cell consists of an Al cathode current collector, an amorphous LiCoO$_2$ (LCO) cathode, a Lipon solid electrolyte, a Si anode, and a Cu anode current collector, as illustrated in Fig. 2a. Amorphous LCO was chosen as the model cathode for this proof-of-concept because it requires no high-

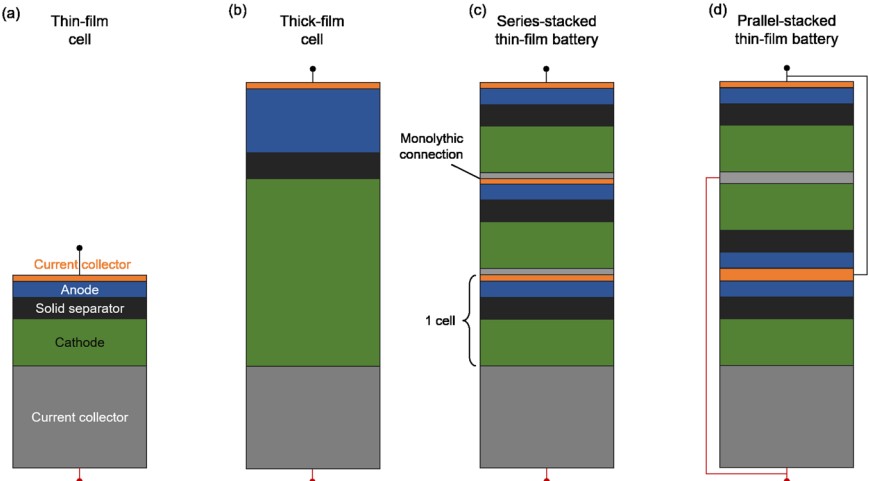

**Fig. 1 Vacuum-deposited solid-state battery designs. a** Thin-film cell consisting of a cathode current collector, cathode, solid electrolyte, anode, and anode current collector. **b** A cell with a thick cathode and anode that are separated by a thin-film solid electrolyte. **c** Series-stacked thin-film battery, whereby several thin-film cells are monolithically stacked and electrically connected in series. **d** Parallel-stacked thin-film battery, whereby several thin-film cells are monolithically stacked and electrically connected in parallel.

**Table 1 Comparison of the different vacuum-deposited battery designs and their properties.**

|  | Thin-film cell | Thick-film cell | Series-stacked thin-film battery | Parallel-stacked thin-film battery |
|---|---|---|---|---|
| Voltage (V) | $1V_O$ | $1V_O$ | $3V_O$ | $1V_O$ |
| Current (A) | $1I_O$ | $4I_O$ | $1I_O$ | $3I_O$ |
| Capacity (Ah) | $1C_O$ | $4C_O$ | $1C_O$ | $3C_O$ |
| Energy (Wh) | $1E_O$ | $4E_O$ | $3E_O$ | $3E_O$ |
| Power (W) | $1P_O$ | $4P_O$ | $3P_O$ | $3P_O$ |
| Resistance (Ω) | $1R_O$ | $4R_O$ | $3R_O$ | $1/3R_O$ |
| Loss (W) | $1P_{loss}$ | $64P_{loss}$ | $3P_{loss}$ | $3P_{loss}$ |
| Energy | $--$ | $++$ | $+$ | $+$ |
| Rate capability | $+$ | $--$ | $+$ | $+$ |
| Thermal management | $+$ | $--$ | $+$ | $+$ |

The energy is a product of voltage and capacity with $E_O = V_O \cdot C_O$. The power is a product of voltage and current with $P_O = V_O \cdot I_O$. The (heat) loss is a function of current and resistance with $P_{loss} = I_O{}^2 \cdot R_O$. We note that we neglect the possible contact resistances of the current collectors and the eventual formation of galvanic corrosion between the two metals.

temperature post-crystallization steps that could otherwise damage the anode and electrolyte. Since monolithic stacking requires a smooth interface between two adjacent cells, we use a Si anode instead of a Li-metal anode, which is often inhomogeneous when deposited via vacuum-based methods. The inhomogeneous deposition of Li metal can lead to contact between the cathode of cell 2 and the anode of cell 1, resulting in battery failure. In contrast, the Si anode is deposited homogeneously, allowing a smooth interface between the two adjacent cells, as seen in the SEM cross-section in Fig. 2b. Figure 2c shows a top view of four series-stacked thin-film batteries. See Supplementary Note 1, Fig. S1, and Fig. S2 for individual electrode performance data, while Supplementary Fig. S3 presents XRD measurements.

The voltage versus capacity curves in Fig. 2d show the charge and discharge behavior of the series-stacked thin-film battery. The voltage of the whole battery corresponds to the sum of the voltages of the two cells, as expected for an electrical series connection of the cells. The average charge and discharge voltages are 7.39 and 6.74 V, respectively. Prior to testing, the cells were precycled individually (see Supplementary Note 2 and Fig. S4).

During charging, current flowed through the entire cell stack, increasing the voltage of both cells. The voltages of cells 1 and 2 were measured independently, and the cycling process was controlled to keep both cells between 3.0 and 4.2 V. Cell 1 had a slightly lower capacity than cell 2. Therefore, cell 1 was the first to reach 4.2 V, while cell 2 remained at a lower state of charge and did not reach 4.2 V. This is more evident at higher $C$-rates (see Supplementary Fig. S5). Figure 2e shows the discharge energy of the series-stacked thin-film battery as a function of the $C$-rate, from $C/10$ to $2C$. The prolonged cycling of the series-stacked thin-film battery for over 300 cycles is shown in Supplementary Fig. S6 in the SI. While the battery's specific energy degrades due to side reactions that reduce the capacity, the battery could operate up to $C$-rates of 2 and for hundreds of cycles without failure. We expect the amorphous LCO to be the limiting factor in the observed cycle life (see also Supplementary Note 1). Supplementary Fig. S7 further shows the discharge capacity of three individual batteries along with their coulombic efficiencies.

**The potential of stacked thin-film batteries**. To further predict the performance potential of stacked thin-film batteries, we used a lumped steady-state thermo-electric model. The model calculates stacked thin-film batteries' specific energy and rate capability, considering the operating limits imposed by voltage efficiency, critical current density, and thermal constraints. The model's concept and basic working mechanism are illustrated in Fig. 3a and described in detail in Supplementary Method 1. The model assumes ten monolithically stacked cells on an Al substrate and lumps the electrochemical behavior of the individual cells into a constant voltage source in series with resistances that represent the ionic and electronic conductivity of the different material layers. The thermal behavior is modeled by Joule heating and 1D steady-state heat conduction (see Supplementary Fig. 8). We assumed anode-free cells, where the Li-metal anode is formed from the Li present in the cathode during the charging of the cell. Such an anode-free thin-film cell has already been achieved using Lipon as the solid electrolyte with critical current densities of up to 5 mA cm$^{-2}$ [31], which can be further increased up to 8 mA cm$^{-2}$ with thin carbon interlayers that are only a few tens of nanometers thick[32].

Figure 3b shows the specific power of the battery as a function of its specific energy in a Ragone plot for the case of a 4 µm thick

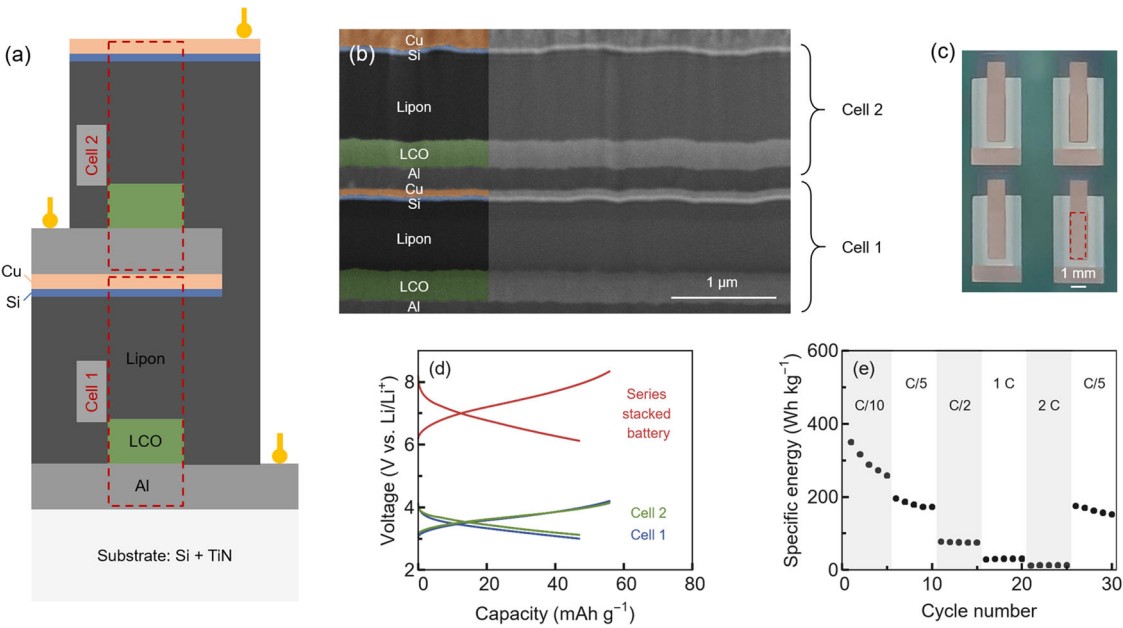

**Fig. 2 Series-stacked thin-film battery. a** Schematic illustration of the fabricated series-stacked thin-film battery. The two cells are marked by red dashed boxes. **b** FIB-SEM cross-section of a monolithically stacked thin-film battery connected in series. The two cells are separated only by thin current collectors, with the cathode current collector of cell 2 directly deposited on the anode current collector of cell 1. **c** Picture of four series-stacked thin-film batteries. The battery area ($1 \times 3$) mm is marked by a red dashed box. **d** Charge–discharge curves of a series-stacked thin-film battery measured at $C/10$ ($1\,\mu A\,cm^{-2}$). The voltage of the series-stacked battery is the combined voltage of the two individual cells, which were cycled simultaneously. **e** The discharge energy of the series-stacked thin-film battery was measured at $C$-rates ranging from $C/10$ ($1\,\mu A\,cm^{-2}$) to $2C$ ($20\,\mu A\,cm^{-2}$).

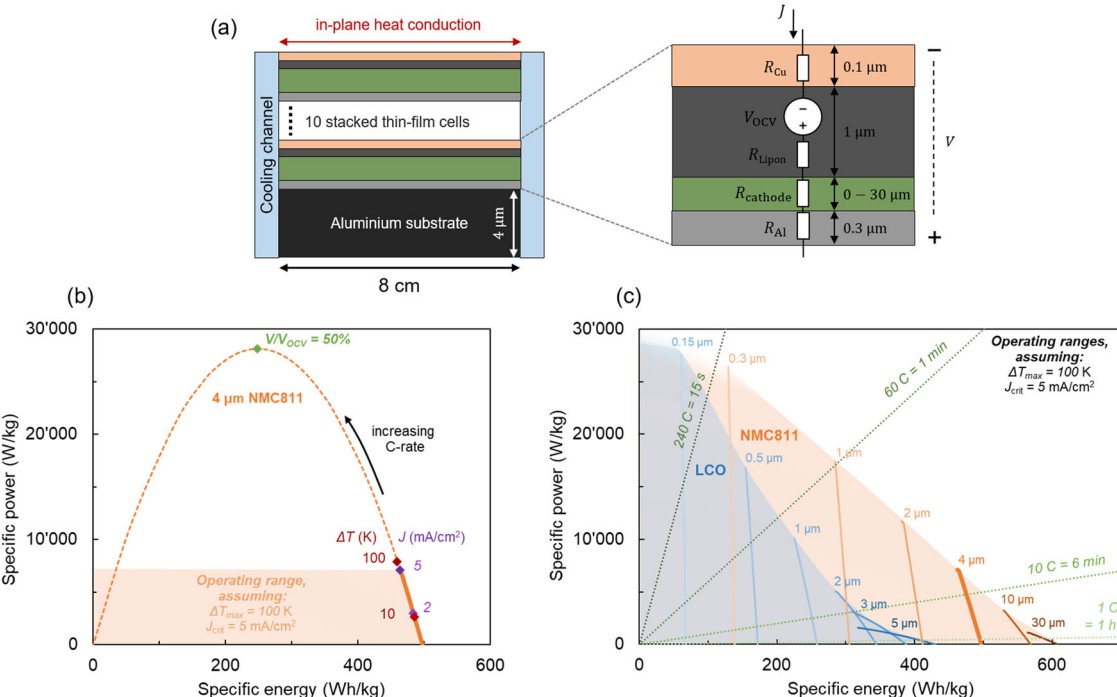

**Fig. 3 Potential of stacked thin-film batteries. a** Schematic illustration of the lumped steady-state thermo-electric model based on a series-stacked thin-film battery with the in-plane heat transfer to cooling channels on the side. Each anode-free cell consists of a 300 nm thick Al cathode current collector, a cathode with variable thickness, a $1\,\mu m$ thick Lipon solid electrolyte, and a 100 nm thick Cu anode current collector. The substrate consists of a $4\,\mu m$ thick Al foil. **b** Ragone plot of a simulated stacked thin-film battery, each cell having a $4\,\mu m$ thick NMC811 cathode. The limiting factors imposed by voltage efficiency ($V/V_{OCV}$), critical current density ($J$), and thermal constraints ($\Delta T$) are indicated. **c** The potential of stacked thin-film batteries is calculated for two different cathode materials—LCO and NMC811—for different cathode thicknesses, which are indicated at the respective lines. The green dashed lines indicate different charging and discharging rates. Note that the specific energies and powers given here are on a stack level, including substrates but not including cell casing.

$LiNi_{0.8}Co_{0.1}Mn_{0.1}O_2$ (NMC811) cathode per cell. Starting from high energy and low power, the power increases with increasing $C$-rate, while the energy decreases due to resistive losses. When the resistive losses lead to a voltage loss of 50%, the power reaches its maximum and decreases at higher $C$-rates (see Supplementary Fig. S9). This decrease in power at high $C$-rates is universal for different battery material systems. Still, it is not usually shown in Ragone plots, as this is not a range of interest for battery operation. The simulated battery consists of ten stacked cells. Further, increasing the number of cells above ten has diminishing returns toward increasing the specific energy (see Supplementary Fig. S10).

In addition to voltage efficiency ($V$), the critical current density ($J$), and the maximum allowable temperature difference between the center of the cell and the cooling channel ($\Delta T$) are two other factors that limit the power of the cell. For the case of 4 μm NMC811 as cathode, these limiting factors are shown in Fig. 3b for a temperature difference of 10 and 100 K and critical current densities of 2 and 5 mA cm$^{-2}$. The limiting factor is different for different cathode thicknesses and materials, as shown in Supplementary Fig. S11.

Figure 3c shows the simulated potential of stacked thin-film batteries for two different cathode materials, LCO and NMC811, with variable thicknesses. For the operating range, we assume a limiting temperature difference of 100 K and a limiting critical current density of 5 mA cm$^{-2}$. For NMC811 cathodes with a thickness larger than 4 μm, specific energies above 500 Wh kg$^{-1}$ are calculated, which coincides with the predicted specific energy limit for SSBs using intercalation cathodes[3]. To obtain high specific power, thin cathodes are required to decrease the resistance losses during fast discharging. Using thin cathodes with a thickness below 1 μm, stacked thin-film batteries can ultimately achieve high specific powers >10 kW kg$^{-1}$ at $C$-rates greater than 60.

**Benchmarking the performance**. To relate the experimental proof-of-concept results and the simulated potential of stacked thin-film batteries to published data on SSBs, we compare our results with literature data for all-SSBs in Fig. 4[11]. Note that our experimental values and literature data include only the anode, solid electrolyte, and cathode and exclude the current collectors and cell casing. The volumetric and gravimetric energy densities, including the current collectors, for three different cells are provided in Supplementary Fig. S12. In accordance with Randau et al.[11], the target performance range of an SSB is specified with an energy of more than 250 Wh kg$^{-1}$ and a cycle rate of more than 1 C, shown as a green-shaded area in Fig. 4. Shown in yellow is the performance of a liquid electrolyte Li-ion battery (LG 18650 HG2L 3000 mAh) with one of the highest reported specific power values for commercially available batteries (including a current collector and cell housing)[33]. While we find that the performance of our series-stacked battery is on par with published results on SSBs, the performances of all the experimental data on SSBs are significantly lower than conventional Li-ion batteries.

The calculated performance of stacked thin-film batteries using LCO and NMC811 cathode is shown in Fig. 4 as a solid blue and orange line, respectively (including current collectors and substrates but without cell casing). It is important to note that all-SSBs, such as the proposed stacked thin-film battery, have significantly lower packaging requirements than conventional Li-ion batteries because of the absence of liquid components. We find that stacked thin-film batteries have the potential to reach specific energies >250 Wh kg$^{-1}$ at a $C$-rate of 10 C for both NMC811 and LCO and even >250 Wh kg$^{-1}$ at a $C$-rate of 60 C for thin NMC811. In comparison, conventional SSBs with an

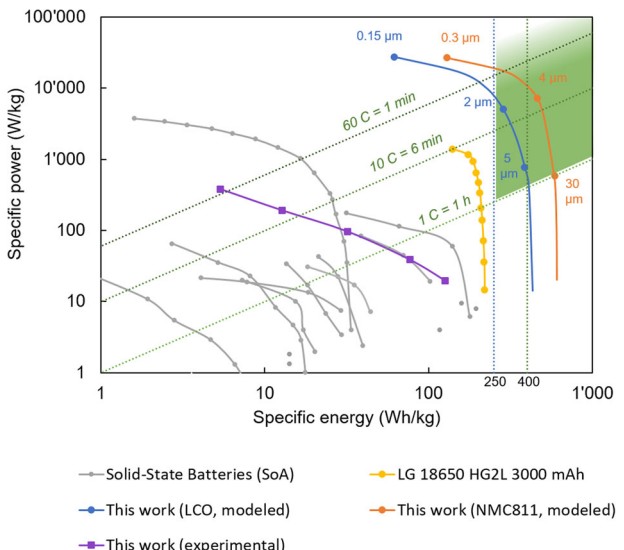

**Fig. 4 Benchmarking the performance of stacked thin-film batteries.** Ragone plot showing the performance of our series-stacked thin-film battery (purple), compared to published results of state-of-the-art (SoA) SSBs measured between 20 and 30 °C (gray)[11]. Our experimental results represent the mean of the values shown in Fig. 2e. Note that both data sets exclude the current collectors and cell casing. Further note that the C rates shown in Fig. 2e refer to initial capacity values under low discharge rates, while the C-rates in Fig. 4 correspond to effective discharge times. The yellow line shows the performance of a commercial lithium-ion battery for power tools, including current collectors and cell casing[33]. Blue and orange lines show the simulated potential of stacked thin-film batteries, including current collectors and substrates for LCO and NMC811 as cathode, respectively. The green shaded area shows the target performance for SSBs.

NMC811 composite cathode thickness of more than 30 μm fail to achieve $C$ rates above 10 C[22]—a distinct difference between stacked thin-film batteries and conventional SSBs.

## Conclusions

We present a high-power and high-energy SSB design based on monolithically-stacked thin-film cells fabricated by scalable vacuum deposition. The individual cells can be electrically connected either in series or in parallel. Although the achievable energy and power of series- and parallel-stacked thin-film batteries are the same, the series-stacked thin-film batteries have some advantages over the parallel-stacked thin-film batteries due to lower currents and simplified cell design. We have experimentally demonstrated a proof-of-concept of a (bipolar) series-stacked thin-film battery. The performance of the fabricated stacked thin-film battery is on par with published results on SSB. We have further predicted the performance potential of stacked thin-film batteries using a thermo-electric model. Our model demonstrates that stacked thin-film batteries can reach specific energies >250 Wh kg$^{-1}$ at $C$-rates greater than 60.

The unique differentiator of thin-film batteries over conventional SSBs is that the diffusion paths are short, which enables high $C$-rates and, thus, high power. While thick (composite) cathodes are well suited for applications requiring high energies, such as batteries for electric vehicles, stacked thin-film batteries with thin cathodes in the order of a few μm are well suited for applications requiring both high power and high energies, such as batteries for drones, robots, or eVTOLs that have stringent requirements for battery performance[34].

There is undoubtedly a long way to go before the full potential of stacked thin-film batteries is realized. Several innovations are

required: (i) Depending on whether high energy or high power is desired, the cathode thickness must be increased to up to tens of µm; (ii) the cathode must be crystallized to maximize capacity but without degradation of the other layers. This may be achieved by rapid thermal annealing[35], photonic annealing methods[36], or the use of cathodes such as vanadates with low annealing temperatures[37]; (iii) the number of stacked cells must ideally be increased up to 10; (iv) the development of stacked thin-film cells in an anode-free design, e.g., by thin seed layers of gold or carbon[32,38]; (v) the fabrication of stacked thin-film batteries on thin substrates, such as thin metal foils[39,40]; (vi) the use of a bipolar current collector to avoid eventual formation of galvanic corrosion between the two metals during cycling.

Open questions remain as to how cycling-related volume changes affect such stacked thin-film batteries, especially when the number of cells increases, and how the vacuum-based deposition methods can be scaled to make stacked thin-film batteries economically viable. The fabrication costs for multi-cell thin-film batteries are expected to be higher than for conventional batteries, as vacuum coating is more expensive in terms of material volume than a slurry coating or printing. Alternative physical vapor deposition processes such as arc deposition and plasma- or thermal spraying can be explored to increase volumetric deposition rates reducing fabrication costs.

## Methods

**Fabrication**. Si wafers with a 110 orientation and a thickness of 525 µm from University wafers were cleaned by ultra-sonication for 15 min, subsequently in detergent in deionized water, deionized water, acetone, and isopropanol. Subsequently, TiN with a thickness of 50 nm was deposited as an adhesion layer using a CT200 magnetron sputtering cluster (Alliance Concept) at a temperature of 450 °C by DC magnetron sputtering of a 25 cm target of Ti at a gas flow of 120 sccm Ar and 10 sccm $N_2$, a power of 3.1 W cm$^{-2}$, and a working pressure of 3 mTorr. All following depositions were performed either by a Nexdep evaporator (Angstrom Engineering Inc.) for thermal evaporation or with an Orion sputtering system (AJA International Inc.) for DC and RF sputtering. First, the cathode current collector Al with a thickness of 300 nm was deposited by thermal evaporation at a rate of 1 Å s$^{-1}$. Next, the LCO cathode with a thickness of 300 nm was deposited at room temperature through a shadow mask ($3 \times 1$ mm) by RF magnetron sputtering of a 2" targets of $Li_2CO_3$ at a gas flow of 24 sccm Ar and 1 sccm $O_2$, a power of 15.3 W cm$^{-2}$, and a working pressure of 3 mTorr. Assuming a density of 4.79 g cm$^{-3}$, this corresponds to an areal mass loading of 0.1437 mg cm$^{-2}$ per cell. The solid electrolyte Lipon with a thickness of 800 nm was deposited at room temperature by RF magnetron sputtering using sputtering of 2" targets of $Li_3PO_4$ at a gas flow of 50 sccm $N_2$, a power of 5.1 W cm$^{-2}$, and a working pressure of 3 mTorr. The Si anode with a thickness of 50 nm was deposited through a shadow mask ($5.5 \times 3$ mm) at room temperature by DC magnetron sputtering using sputtering of 2" targets of Si at a gas flow of 24 sccm Ar, a power of 3.0 W cm$^{-2}$, and a working pressure of 4 mTorr. To finish the first cell, the anode current collector Cu with a thickness of 100 nm was deposited through a shadow mask ($5.5 \times 3$ mm) by thermal evaporation at a rate of 1 Å s$^{-1}$. Subsequently, to deposit the second cell on top of the first cell, Al, LCO, Lipon, Si, and Cu were deposited as described above through different shadow masks (Al: $5.5 \times 3$ mm, LCO: $3 \times 1$ mm, Lipon: $6 \times 2$ mm, Si and Cu: $5 \times 1$ mm).

**Characterization**. Cross-section SEM images were obtained with a Helios Nano-Lab 600 DualBeam system. Transfer to the FIB-SEM was performed in air with an ambient time of less than 30 seconds. The cross-section was milled with an ion beam current of 0.77 nA, followed by a cleaning cut with an ion beam current of 80 pA at 30 kV. Electrochemical characterization was performed in an Ar-filled glovebox at room temperature without applied pressure using a Squidstat potentiostat (Admiral Instruments). The reported capacities correspond to electrode-level capacities. The theoretical capacity for amorphous LCO used to calculate the C-rate was assumed to be 70 mAh g$^{-1}$.

**Modeling**. The performance potential of stacked thin-film SSBs was modeled using a lumped steady-state thermo-electric model, the details of which are described in Supplementary Method 1.

## Data availability

The data that support the findings of this study are available from the corresponding author upon reasonable request.

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

## Acknowledgements

The authors thank Ayodhya N. Tiwari and Maksym V. Kovalenko for carefully reading and commenting on the manuscript. The work is supported by the Strategic Focus Area (SFA) Advanced Manufacturing of the ETH Domain (project "SOL4BAT") and the Swiss National Science Foundation (grant number 200021_172764). M.H.F. is supported by a Rubicon Fellowship from the Netherlands Organization for Scientific Research (NWO).

## Author contributions

M.H.F. conceived the experimental and theoretical work, designed and carried out experimental work, and prepared the paper; L.B. designed and carried out experimental and theoretical work and prepared the paper. A.M. carried out experimental work and performed FIB-SEM measurements. J.C. carried out experimental work and performed XRD measurements. A.A. assisted in thin-film fabrication and helped design experimental and theoretical work. Y.E.R. conceived and supervised the project. All authors reviewed and commented on the paper.

## Competing interests

M.H.F., A.A., and Y.E.R. are founders of BTRY AG, a company commercializing solid-state batteries. The remaining authors declare no competing interests.
