## [Peer Review File · Communications Chemistry]

nature portfolio

Peer Review File

Monolithically-stacked thin-film solid-state batteriesReviewers' comments:

Reviewer #1 (Remarks to the Author):

The article entitled "Monolithically-stacked thin-film solid-state batteries" is a very interesting report highlighting the importance of thin-film technology in solid state Li-ion batteries. However, there are several aspects of this work that should be deeply revisited. For this reason, at this stage, I cannot recommend the publication of this work.

The chosen electrode couple is Si / amorphous LCO deposited by DC and RF magnetron sputtering respectively. There has not been performed any structural or compositional characterization that corroborates the existence of the desired electrode materials. How do you know that LCO has the target stoichiometry? And what about Si? Is it amorphous? How do you know it is free of SiO₂?

The electrochemical characterization is not sufficient. The curves shown do not support the existence of the expected Li-ion insertion, there are not clear insertion/deinsertion plateau. In this case, it is necessary to show a dQ/dV vs voltage curve that will help to understand the potential of the redox reactions and also how many of them are taking place. Without this information is impossible to assess the existence of a storage mechanism and its capacity.

In Figure 1e, the resistance of the series-stacked thin-film battery is determined to be $3xR_0$. However, in this case the contact resistance of anode and cathode current collectors are being neglected. The authors should explain this approximation to zero-resistance and also the eventual formation of galvanic corrosion between these two metals.

In the introduction, the relevance of references 18 and 19 is discussed. Indeed, the works of this group are pioneering the application of thin-film technology to Li-ion batteries. However, the scale up of their concepts is limited since the substrates used for the electrode deposition by means of RF magnetron sputtering are precious metals such as platinum. In contrast, recent studies (ACS Appl. Energy Mater. 5, 12120–12131) have shown the possibility to grow monolithic thin-film solid-state batteries on standard stainless-steel substrates by means of AC magnetron sputtering which, in addition, displays higher deposition rates than RF magnetron sputtering.

Si anode is used instead of Li anode due to the inhomogeneous deposition of the latter. However, Si is known for the massive volume expansion upon Li-ion insertion (up to 400% expansion) which would lead into rapid degradation of the Si anode and problems at the interface between two adjacent cells. The authors should comment on this and explain how the volume expansion has been eventually bypassed.

The limited cycle life is ascribed to the amorphous LCO, however, other effects such as volume expansion in Si could be at the origin of the limited cycle life.

The authors present the cell stacking as a concept "also used in conventional SSBs to simplify cell design by reducing external connections and cooling system requirements". Indeed, this has been used in conventional SSBs, however, the cell stacking in thin film cells is something already studied and presented in several patents:

- Monolithically integrated thin-film solid state lithium battery device having multiple layers of lithium electrochemical cells, US20120058380A1.
- Thin film battery module, thin film battery package, thin film battery package manufacturing device, and thin film battery package manufacturing method, US20140370364A1.

The difference between the current work and previous publications must be explained.

Although the research topic is of great interest and the results have a good potential, the points above should be addressed before considering resubmission.

Reviewer #2 (Remarks to the Author):

This paper details the development and proof of concept of stacked thin film batteries based on the Lipon thin film battery. Overall I think this is a nice and valuable piece of work but some comments should be addressed.

1) The authors oversell the work some. They emphasize that only thin cathodes can provide the power they are looking for. While high power is important the compromise in energy density is pretty critical parameter as well.

2) While vapor deposition does have significant advantages, it should be clearly noted the potential cost vs. other technologies. The biggest road block to Lipon based batteries is not power or energy, but rather the cost of manufacturing and the difficulty in production. The sputtering of metal oxides in particular is very expensive.

Overall thought the demonstration of stacked thin film batteries is of value. I am particularly interested in further research showing how the volume change affects the long term performance.

Reviewer #3 (Remarks to the Author):

Futscher and co-authors present in their manuscript a lithium-ion battery design, where single thin films cells are stacked monolithically. They present results on a series-stacked thin film battery and demonstrate that this kind of stacking increases the voltage, i.e. the voltage of the stacked design corresponds to the sum of the voltage of both individual cells. In addition, they perform simulations showing that high specific powers can be achieved theoretically. It is not clear, what new design is reported by the authors, as the design of series-stacked thin film batteries is already reported and demonstrated in literature, see e.g. Kong et al. *Chem. Comm.* 57, 12587; Schnell et al., *J. Power Sources* 382, 160; Y.-S. Hu, *Nat. Energy*, 1, 16042; Kim et al., *J. MaterChem. A* 2, 10862. As I do not see any significant difference compared to previous reports, the work doesn't provide any new design ideas or insights. Thus, I cannot recommend the manuscript for publications.

We would like to thank the reviewers for taking the time to evaluate our manuscript. We have provided a point-to-point response to each of the reviewer's comments below. The original reviewer's comments are in black, and our response is in green. The revised manuscript, with changes highlighted, is attached to this response to the reviewers.

Reviewer #1 (Remarks to the Author):

The article entitled "Monolithically-stacked thin-film solid-state batteries" is a very interesting report highlighting the importance of thin-film technology in solid state Li-ion batteries. However, there are several aspects of this work that should be deeply revisited. For this reason, at this stage, I cannot recommend the publication of this work.

We thank the reviewer for constructive criticism of our work. To address the issues raised, we fabricated new electrode layers for structural characterisation and new single cells to evaluate the electrochemical performance of the individual electrodes. The answers to the reviewer's questions are given below and are included in the revised manuscript.

The chosen electrode couple is Si / amorphous LCO deposited by DC and RF magnetron sputtering respectively. There has not been performed any structural or compositional characterization that corroborates the existence of the desired electrode materials. How do you know that LCO has the target stoichiometry? And what about Si? Is it amorphous? How do you know it is free of SiO₂?

We acknowledge the limitations of our work, especially considering the performance of the electrodes used in our proof-of-concept battery. To address and overcome these limitations, we have discussed various approaches in the Outlook section of the manuscript. Since the electrodes are not yet optimised for multi-cell batteries, we have focused on the cell design, upcoming challenges, and potential in the manuscript while limiting the characterisation of the electrodes. However, to further verify their amorphous nature, we have measured XRD of the individual electrodes and added the measurements to the revised manuscript's supplementary information, which now reads:

The Bragg-Brentano XRD diffractograms presented in Figure S3a confirm the absence of any crystalline phase in both as-deposited Si and LCO electrodes. Furthermore, grazing-incidence XRD measurements of LCO (Figure S2b) and Si (Figure S2c) also show no evidence of the LCO (003) and Si (111) peaks, further verifying that no crystalline phase is present for the LCO cathode or the Si anode.

Figure S3a. (a) Bragg-Brentano XRD diffractograms of as-deposited 300 nm thick Si and LCO on alkaline aluminoborosilicate glass substrates (Corning EXG). (b) Grazing-incidence XRD (GI-XRD) of as-deposited 300 nm thick LCO at an incident angle of 1° in the two theta angle range of 18.5° to 19.5°. (c) GI-XRD of as-deposited 300 nm thick Si at an incident angle of 1.5° in the two theta angle range of 28° to 29°.

The electrochemical characterization is not sufficient. The curves shown do not support the existence of the expected Li-ion insertion, there are not clear insertion/deinsertion plateau. In this case, it is necessary to show a dQ/dV vs voltage curve that will help to understand the potential of the redox reactions and also how many of them are taking place. Without this information is impossible to assess the existence of a storage mechanism and its capacity.

To better understand the storage mechanism and its capacity, we have measured differential capacity profiles and added them to the SI of the revised manuscript, which now reads:

Figure S2 shows the individual electrodes' differential capacity (dQ/dV) profiles in a single-cell configuration with a Li-metal anode and the multi-cell battery, corresponding to the first five cycles at C/10. The LCO cell exhibits one insertion and desorption contribution each, and the position of these redox reactions are consistent with expectations for LCO, but broadened due to the amorphous nature of the films (Figure S2a). In addition, the fast capacity fade is clearly visible. Si demonstrates two insertion and desorption contributions: the insertion reaction of $a\text{-Si} \rightarrow a\text{-Li}_{2.0}\text{Si} \rightarrow a\text{-Li}_{3.5}\text{Si}$ and the desinsertion reaction of $a\text{-Li}_{3.5}\text{Si} \rightarrow a\text{-Li}_{2.0}\text{Si} \rightarrow a\text{-Si}$ (Figure S2b). The differential capacity profile of the multi-cell battery features broad insertion and desorption profiles indicative of LCO, superimposed with peaks that can be assigned to the insertion and desinsertion profiles of Si (Figure S2c).

Figure S2. Differential capacity profiles of (a) LCO/Lipon/Li cell, (b) Si/Lipon/Li cell, and (c) multi-cell battery.

In Figure 1e, the resistance of the series-stacked thin-film battery is determined to be $3 \times R_0$. However, in this case the contact resistance of anode and cathode current collectors are being neglected. The authors should explain this approximation to zero-resistance and also the eventual formation of galvanic corrosion between these two metals.

We thank the reviewer for bringing this to our attention. We acknowledge that our study did not account for any contact resistances between the anode and cathode current collectors. We have added a new sentence to the manuscript, which reads as follows: "We note that we neglected the possible contact resistances of the current collectors and the eventual formation of galvanic corrosion between the two metals." Additionally, we have included a mention of the need for a bipolar current collector in the Outlook section of our paper, which now reads as follows: "Several innovations are required: ... vi) The use of a bipolar current collector to avoid eventual formation of galvanic corrosion between the two metals during cycling."

In the introduction, the relevance of references 18 and 19 is discussed. Indeed, the works of this group are pioneering the application of thin-film technology to Li-ion batteries. However, the scale up of their concepts is limited since the substrates used for the electrode deposition by means of RF magnetron sputtering are precious metals such as platinum. In contrast, recent studies (ACS Appl. Energy Mater. 5, 12120–12131) have shown the possibility to grow monolithic thin-film solid-state batteries on standard stainless-steel substrates by means of AC magnetron sputtering which, in addition, displays higher deposition rates than RF magnetron sputtering.

We thank the reviewer for pointing out our oversight in mentioning the need for thin substrates. We agree that using thick and conventional substrates, such as Si-wafers or sapphire wafers, is detrimental to the energy density of thin-film solid-state batteries. Our research group has also demonstrated the feasibility of fabricating thin-film solid-state batteries on flexible Al foil (<https://doi.org/10.1021/acsaem.1c01283>).

We have acknowledged this in the revised Outlook section of our manuscript, which now states: "Several innovations are required: v) The fabrication of stacked thin-film batteries on thin substrates, such as thin metal foils ". We have also included the reference kindly provided by the reviewer to support this statement.

The need for different fabrication methods with higher volumetric deposition rates is further acknowledged in the Outlook section: "Alternative physical vapour deposition processes such as arc deposition and plasma- or thermal spraying can be explored to increase volumetric deposition rates reducing fabrication costs."

Si anode is used instead of Li anode due to the inhomogeneous deposition of the latter. However, Si is known for the massive volume expansion upon Li-ion insertion (up to 400% expansion) which would lead into rapid degradation of the Si anode and problems at the interface between two adjacent cells. The authors should comment on this and explain how the volume expansion has been eventually bypassed.

The limited cycle life is ascribed to the amorphous LCO, however, other effects such as volume expansion in Si could be at the origin of the limited cycle life.

We acknowledge the reviewer's concerns regarding the impact of Si's significant volume expansion on the durability of multi-cell batteries, specifically as the cathode can store more energy. However, in our case, the cathode loading per cell ($4 \mu\text{Ah}/\text{cm}^2$) is considerably lower than the anode loading ($41 \mu\text{Ah}/\text{cm}^2$). As a result, we expect to observe only a small volume change of about 10 nm within the Si anode.

We further conducted cycling measurements of the individual electrodes to identify the underlying cause of the rapid degradation observed in our proof-of-concept cell. The LCO cathode exhibited a substantial capacity fade within the first five cycles of 21% (Figure S1d), compared to 5% of the Si anode (Figure S1e). On the other hand, the multi-cell battery showed a capacity fade of 26% within the first 5 cycles (Figure S1f). Therefore, we conclude that the capacity fade of the LCO electrode is responsible for most of the capacity fade observed in our multi-cell battery. We added the information to the SI of the revised manuscript.

Figure S1. Charge-discharge curves and discharge capacity of (a) and (d) LCO/Lipon/Li cell, (b) and (e) Si/Lipon/Li cell, and (c) and (f) multi-cell battery, respectively. The C-rates are calculated assuming a capacity of $70 \text{ mAh}\cdot\text{g}^{-1}$ for LCO and $4.2 \text{ Ah}\cdot\text{g}^{-1}$ for Si, resulting in a current density of 10, 49, and $10 \mu\text{A}\cdot\text{cm}^{-2}$ for the LCO cell, the Si cell, and the multi-cell battery at 1 C, respectively.

The significance of volume expansion on the lifetime of multi-cell batteries is further discussed in the Outlook section of our manuscript, which reads: "Open questions remain as to how cycling-related volume changes affect such stacked thin-film batteries, especially when the number of cells increases, and how the vacuum-based deposition methods can be scaled to make stacked thin-film batteries economically viable."

The authors present the cell stacking as a concept "also used in conventional SSBs to simplify cell design by reducing external connections and cooling system requirements". Indeed, this has been used in conventional SSBs, however, the cell stacking in thin film cells is something already studied and presented in several patents:

- Monolithically integrated thin-film solid state lithium battery device having multiple layers of lithium electrochemical cells, US20120058380A1.

- Thin film battery module, thin film battery package, thin film battery package manufacturing device, and thin film battery package manufacturing method, US20140370364A1.

The difference between the current work and previous publications must be explained.

We agree with the reviewer's observation that stacking thin-film cells is not a new concept, as it has been previously discussed in patents. There have been industrial efforts to commercialise stacked thin-film cells, such as by Sakti3. However, to our knowledge, there has been no reported demonstration of a working stacked thin-film battery within both patents and academic literature, except for a study published in 2003 which demonstrated only one cycle at a C-rate of C/3 that we just found very recently. Our work demonstrates stacked thin-film batteries over many cycles with C-rates up to 60. In addition, our thermoelectric model provides valuable insights into the potential performance of multi-cell thin-film batteries, which, to the best of our knowledge, has not yet been reported in either patents or academic literature and is unfamiliar to many battery researchers.

We have added the following statement in the manuscript: "Stacked thin-film batteries have further been discussed in patents and previously attempted to be commercialised by a company called Sakti3. However, to our knowledge, there has been no reported demonstration of a working stacked thin-film battery within both patents and academic literature, except for a study published in 2003 which demonstrated only one cycle at a C-rate of C/3. Our work demonstrates the stacked thin-film batteries over many cycles with C-rates up to 60.

Although the research topic is of great interest and the results have a good potential, the points above should be addressed before considering resubmission.

Reviewer #2 (Remarks to the Author):

This paper details the development and proof of concept of stacked thin film batteries based on the Lipon thin film battery. Overall I think this is a nice and valuable piece of work but some comments should be addressed.

We thank the reviewer for the positive assessment of our work. We have addressed the comments below.

1) The authors oversell the work some. They emphasize that only thin cathodes can provide the power they are looking for. While high power is important the compromise in energy density is pretty critical parameter as well.

We fully agree with the reviewer that energy and power density are crucial parameters. The reviewer may have overlooked the central message of our work. Our work demonstrates that thin-film cathodes can offer similar energy densities as conventional cell designs when stacked monolithically while at the same time providing a much better rate capability. This key finding is emphasised throughout the manuscript, for example, in the abstract and conclusion and is clearly illustrated in the figures.

2) While vapor deposition does have significant advantages, it should be clearly noted the potential cost vs. other technologies. The biggest road block to Lipon based batteries is not power or energy, but rather the cost of manufacturing and the difficulty in production. The sputtering of metal oxides in particular is very expensive.

The reviewer's observation is accurate. Vacuum processing is more expensive than conventional liquid processing. We have already acknowledged this in the Outlook section of our paper, which states: "The fabrication costs for multi-cell thin-film batteries are expected to be considerably higher than for conventional batteries, as vacuum coating is more expensive in terms of material volume than a slurry coating or printing. Alternative physical vapour deposition processes such as arc deposition and plasma- or thermal spraying can be explored to increase volumetric deposition rates reducing fabrication costs."

Although not included in the paper, we conducted cost estimations internally for a fully vacuum-processed battery. Our findings indicate that such a battery could potentially cost approximately 100 times more than conventional batteries (10'000 vs 100 \$/kWh) but only 2 times more than supercapacitors (5'000 \$/kWh). Despite the cost difference, we believe that the benefits of having high energy and power densities justify the practical

price for high-end applications. However, such a techno-economic analysis would be beyond the scope of this work.

Overall thought the demonstration of stacked thin film batteries is of value. I am particularly interested in further research showing how the volume change affects the long term performance.

We thank the reviewer for acknowledging the value of our work. While the long-term effects of volume changes on cycling performance are beyond the scope of this study, we acknowledge this issue in the Outlook section, which reads: "Open questions remain as to how cycling-related volume changes affect such stacked thin-film batteries, especially when the number of cells increases". Future work will aim to address this question. We hope to investigate this further in future studies.

Reviewer #3 (Remarks to the Author):

Futscher and co-authors present in their manuscript a lithium-ion battery design, where single thin films cells are stacked monolithically. They present results on a series-stacked thin film battery and demonstrate that this kind of stacking increases the voltage, i.e. the voltage of the stacked design corresponds to the sum of the voltage of both individual cells. In addition, they perform simulations showing that high specific powers can be achieved theoretically.

It is not clear, what new design is reported by the authors, as the design of series-stacked thin film batteries is already reported and demonstrated in literature, see e.g. Kong et al. Chem. Comm. 57, 12587; Schnell et al., J. Power Sources 382, 160; Y.-S. Hu, Nat. Energy, 1, 16042; Kim et al., J. MaterChem. A 2, 10862. As I do not see any significant difference compared to previous reports, the work doesn't provide any new design ideas or insights. Thus, I cannot recommend the manuscript for publications.

We thank for the time spent by the reviewer to evaluate our manuscript.

We agree with the reviewer's observation that stacking alone is not a novel concept, and we have already acknowledged in our manuscript that stacked batteries obtained by mechanical stacking are well known: "We note that stacking cells is a concept also used in conventional SSBs to simplify cell design by reducing external connections and cooling system requirements. While bipolar stacked SSBs promise a 30 % increase in specific energy and power, their rate capability would still be limited by heat generation within the electrodes and solid-state separators.". While some of the references we had already cited within the original manuscript, we have further incorporated some of the references suggested by the reviewer in the revised manuscript.

However, we would like to note that the kindly provided references all discuss bulk cells using composite cathodes, which are fundamentally different from the thin-film cathodes, as discussed in our manuscript. While bulk cells can achieve higher energy densities, they are still fundamentally limited to low charging rates governed by composite cathodes. Our paper highlights the advantages of using thin-film cathodes, such as high-rate capability, to construct solid-state batteries with high energy and power densities. To achieve this, PVD methods are necessary to produce separators and current collectors with thicknesses below 1 μm , which is not feasible with conventional fabrication methods. Additionally, we have developed a theoretical model and presented its results, demonstrating the potential of such multi-cell batteries, which, to the best of our knowledge, has not yet been reported in either patents or academic literature and is unfamiliar to many battery researchers. Due to these reasons, we still strongly consider our results as novel.

Reviewers' comments:

Reviewer #1 (Remarks to the Author):

The authors have addressed all the points raised except one while providing new experimental data. I would like to acknowledge their effort. Unfortunately, the electrochemical couples and their performance is far from good, as a result, I am still reluctant to accept the paper in its current form. There are some points that should be addressed/changed before this paper can be accepted.

- The question that I made in the previous revision about the measured stoichiometry of the electrode materials has not been addressed. There is no evidence of the stoichiometry of the LCO or LiPON, nor data about Si purity. This should be clarified in the paper, clearly indicating that the lithium cobalt oxide and the lithium phosphorous oxynitride have unknown stoichiometry and the purity of the Si is not known. For this reason, the materials should not be called LCO or LiPON.

- Moreover, the so called LCO and LiPON are amorphous. This means that they will never have the electrochemical response expected. In fact, the dQ/dV of LCO should display two peaks between 3.9 and 4.0 V (J. Power Sources 196, 697)

- The electrochemical response for Si that the authors have presented would match the expected behavior (the fact that Si is amorphous is helping here). However, their explanation about the electrochemical reaction is wrong: the Li on Si does not intercalate!!! This system is working following an alloying reaction. This is a considerable error that should be amended.

- In my opinion the title of the paper should reflect that the authors do not have conventional electrochemical couples. I understand that the stacking concept is interesting, however by using the term solid-state battery one would expect having a set of electroactive materials that work as expected e.g. crystalline LCO with the proper electrochemical response.

If the authors correct and modify the manuscript to consider the above points I would be willing to support the publication of the manuscript.

We have provided a point-to-point response to each of the reviewer's comments below. The original reviewer's comments are in black, and our response is in green.

Reviewer #1 (Remarks to the Author):

The authors have addressed all the points raised except one while providing new experimental data. I would like to acknowledge their effort. Unfortunately, the electrochemical couples and their performance is far from good, as a result, I am still reluctant to accept the paper in its current form. There are some points that should be addressed/changed before this paper can be accepted.

Thank you for taking the time to evaluate our manuscript and acknowledging the effort we have made to address the concerns raised.

- The question that I made in the previous revision about the measured stoichiometry of the electrode materials has not been addressed. There is no evidence of the stoichiometry of the LCO or LiPON, nor data about Si purity. This should be clarified in the paper, clearly indicating that the lithium cobalt oxide and the lithium phosphorous oxynitride have unknown stoichiometry and the purity of the Si is not known. For this reason, the materials should not be called LCO or LiPON.

- Moreover, the so called LCO and LiPON are amorphous. This means that they will never have the electrochemical response expected. In fact, the dQ/dV of LCO should display two peaks between 3.9 and 4.0 V (J. Power Sources 196, 697)

LiPON is a standard material in our laboratory and within the thin-film battery community. It is well-known to be amorphous, dating back to studies from the 1990s from the Oak Ridge National Laboratories. We process LiPON as described in detail in the materials and methods section. Its stoichiometry and its electrochemical response are known and well-established, although it is not explicitly presented in this paper, as we do not see any added benefit to this work.

Regarding the amorphous LCO, we agree that its stoichiometry is not known. However, since we know and discuss the shortcomings of amorphous LCO compared to crystalline LCO, we do not see any added benefit in further investigating its stoichiometry. The abbreviation "LCO" is used because the elements present in the electrode remain the same as those in crystalline LCO. To clearly distinguish our material from the crystalline form, we have already referred to it as "amorphous LCO" throughout the manuscript. We now also mention that the stoichiometry of amorphous LCO is unknown in the SI.

We acknowledge that the dQ/dV for the amorphous LCO differs from the ones for crystalline LCO. We have further emphasised this point in the SI. In our previous publications, when we crystallise our films, we obtain the expected stoichiometry for crystalline LCO and the expected insertion and desorption plateaus, as shown in, e.g., 10.1149/1945-7111/abf215 and 10.1021/acsami.0c09777. However, due to the multi-cell stack configuration in our current study, thermal annealing, which is required to obtain crystalline LCO, is unfortunately not possible. As a result, we have to use amorphous LCO in this work. In the manuscript, we have mentioned the necessary steps for further improvements in the outlook, including the use of alternative crystallisation methods.

- The electrochemical response for Si that the authors have presented would match the expected behavior (the fact that Si is amorphous is helping here). However, their explanation about the electrochemical reaction is wrong: the Li on Si does not intercalate!!! This system is working following an alloying reaction. This is a considerable error that should be amended.

We appreciate the reviewer's observation and would like to clarify that our manuscript does not state otherwise. In fact, we have provided the alloying reactions in the SI. We have further emphasised this point in the SI to avoid any confusion, mentioning explicitly that it is indeed alloying and dealloying reactions.

- In my opinion the title of the paper should reflect that the authors do not have conventional electrochemical couples. I understand that the stacking concept is interesting, however by using the term solid-state battery one

would expect having a set of electroactive materials that work as expected e.g. crystalline LCO with the proper electrochemical response.

We believe that our current title is still appropriate for the following reasons: i) Our work focuses on a solid-state battery system, as per the definition of such systems. In fact, compared to many other studies that present single cells and refer to them as batteries, our study genuinely encompasses a battery, as it contains more than one cell. ii) In addition to presenting a proof-of-concept battery for the stacking approach, our paper also delves into the development of a model of monolithically-stacked thin-film solid-state batteries to demonstrate their potential. The current title encompasses both the stacking concept and the model's purpose, while a revised title focusing solely on the unconventional electrochemical couples would not capture the scope of our work.

If the authors correct and modify the manuscript to consider the above points I would be willing to support the publication of the manuscript.

We appreciate the reviewer's support for the publication of our manuscript.